# Role of Ranolazine in the Prevention and Treatment of Atrial Fibrillation in Patients with Left Ventricular Systolic Dysfunction: A Meta-Analysis of Randomized Clinical Trials

**DOI:** 10.3390/diseases9020031

**Published:** 2021-04-16

**Authors:** Pattranee Leelapatana, Charat Thongprayoon, Narut Prasitlumkum, Saraschandra Vallabhajosyula, Wisit Cheungpasitporn, Ronpichai Chokesuwattanaskul

**Affiliations:** 1Cardiac Center, Division of Cardiovascular Medicine, Department of Internal Medicine, Faculty of Medicine, Chulalongkorn University and King Chulalongkorn Memorial Hospital, Bangkok 10330, Thailand; 2Division of Nephrology and Hypertension, Mayo Clinic, Rochester, MN 55905, USA; charat.thongprayoon@gmail.com; 3Division of Cardiology, University of California Riverside, Riverside, CA 92521, USA; narutprasitlumkum@gmail.com; 4Section of Interventional Cardiology, Division of Cardiovascular Medicine, Department of Medicine, Emory University School of Medicine, Atlanta, GA 30322, USA; saraschandra.vallabhajosyula@emory.edu

**Keywords:** atrial fibrillation, ranolazine, systolic dysfunction, heart failure, meta-analysis

## Abstract

Background: Ranolazine has the potential to prevent atrial fibrillation (AF) and plays a role in rhythm control strategy for atrial fibrillation in various clinical settings. However, data on the use of ranolazine in patients with left ventricular (LV) systolic dysfunction are limited. The aims of this meta-analysis of randomized clinical trials are to investigate the efficacy and safety of ranolazine in AF patients with LV systolic dysfunction. PubMed and the Cochrane Database of Systematic Reviews were searched until July 2020. The efficacy outcomes included the incidence of new-onset AF, the rate of sinus rhythm restoration, and the time until sinus rhythm restoration. Safety endpoints were at death, and any adverse events were reported in the enrolled studies. We initially identified 204 studies and finally retrieved 5 RCTs. Three studies were analyzed in the meta-analysis. Among AF patients with LV systolic dysfunction, our meta-analysis showed that the combination of ranolazine to amiodarone significantly increased the sinus rhythm restoration rate compared to amiodarone alone (risk ratio (RR) 2.87, 95% confidence interval (CI) 2.48–3.32). Moreover, the time to sinus rhythm restoration was 2.46 h shorter in the ranolazine added to amiodarone group (95% CI: 2.27–2.64). No significant adverse events and proarrhythmias in the ranolazine group were identified. In conclusion, in AF patients with LV systolic dysfunction, ranolazine as an add-on therapy to amiodarone potentiates and accelerates the conversion of AF to sinus rhythm. Moreover, ranolazine shows good safety profiles. Further studies to investigate the effectiveness of ranolazine in the prevention of new-onset AF among patients with LV systolic dysfunction are needed.

## 1. Introduction

For patients with atrial fibrillation (AF) and left ventricular (LV) systolic dysfunction, most of the recent studies favor rhythm control, especially by catheter ablation over rate control [1,2,3]. Pulmonary vein isolation is an effective and safe AF ablation technique that has been approved as the intervention of choice for restoring and maintaining sinus rhythm [4,5]. However, despite the rapid advancement of AF ablation technologies, the recurrence rate of AF following catheter ablation is still high and is estimated to be 20–45% [1,2,6,7,8,9]. For this reason, when being used either as a stand-alone or add-on therapy to catheter ablation, antiarrhythmic drugs are still being considered as a cornerstone of treatment. Unfortunately, most antiarrhythmic drugs have adverse effects, and only a few can be safely used among patients with LV systolic dysfunction [3].

Ranolazine selectively inhibits the atrial peak sodium channel current (INa), resulting in increased atrial post-repolarization refractoriness, which may account for AF suppression [10,11]. Previous studies support this hypothesis, demonstrating that ranolazine has the potential to prevent AF and convert it to sinus rhythm in various clinical settings [12,13,14]. However, the data on its efficacy and safety among AF patients with LV systolic dysfunction are still lacking. Therefore, the purpose of this study was to conduct a comprehensive meta-analysis of all published randomized controlled trials (RCTs) that investigated the efficacy and safety of ranolazine, either being used as a single drug or as an adjunctive therapy in patients with AF and LV systolic dysfunction.

## 2. Materials and Methods

### 2.1. Literature Review and Search Strategy

The protocol for this systematic review and meta-analysis was registered with PROSPERO (International Prospective Register of Systematic Reviews): CRD42020204769. A systematic literature search of PubMed (2004 to July 2020) and the Cochrane Database of Systematic Reviews (database inception to July 2020) was conducted to investigate the pooled sinus rhythm conversion in AF in patients with LV systolic dysfunction and the time to sinus rhythm restoration in AF patients with LV systolic dysfunction.

The systematic literature review was undertaken independently by two investigators (R.C. and P.L.) applying a search approach that incorporated the terms of “atrial fibrillation” OR “AF” OR “AFib” combined with the term “ranolazine” and the term “LV systolic dysfunction” OR “systolic heart failure” OR “heart failure reduced ejection fraction” OR “heart failure reduced EF” (Appendix A). A manual search for conceivably relevant studies using references of the included articles was also performed. No language limitation was applied. This study was conducted following the PRISMA (Preferred Reporting Items for Systematic Reviews and Meta-Analysis) statement (Appendix A) [15]. The raw data for this systematic review is publicly available through the Open Science Framework (URL: osf.io/2vuyg (accessed on 8 September 2020)).

### 2.2. Selection Criteria

Eligible studies had to be randomized studies that reported incidence of AF, the time to sinus rhythm restoration, or the sinus rhythm conversion rate in AF patients with LV systolic dysfunction. Inclusion was not limited by study size. The acquired articles were independently reviewed for their eligibility by the two investigators noted previously. Any discrepancies were discussed and resolved by mutual consensus.

### 2.3. Data Extraction

A structured data collecting form was utilized to derive the following information from each study, including: title, year of the study, name of the first author, publication year, country where the study was conducted, demographic and characteristic data of AF patients, methods used to identify AF, time to sinus rhythm restoration, and sinus rhythm restoration rate of AF patients with LV systolic dysfunction.

### 2.4. Quality Assessment

Two blinded reviewers (P.L., R.C.) assessed the risk of bias by the Cochrane Collaboration’s tool for assessing the risk of bias in randomized trials. The following risks of bias were evaluated: random sequence generation, allocation concealment, blinding of participants and personnel, blinding of outcome assessment, incomplete outcome data, and other bias.

### 2.5. Statistical Analysis

Analyses were performed using R software, version 3.6.3 (R Foundation for Statistical Computing, Vienna, Austria). Adjusted point estimates from each study were consolidated by the generic inverse variance approach of DerSimonian and Laird, which designated the weight of each study based on its variance [16]. Cochran’s Q test and I^2^ statistics were applied to determine the between-study heterogeneity. Given the possibility of a between-study variance, we used a random-effects model rather than a fixed-effects model. A value of I^2^ of 0–25% indicates insignificant heterogeneity, 26–50% low heterogeneity, 51–75% moderate heterogeneity, and 76–100% high heterogeneity [17]. The presence of publication bias was evaluated via the Egger test [18].

## 3. Results

A total of 204 potentially eligible articles were identified using our search strategy. After the exclusion of 17 duplicate articles and 150 articles, because they were case reports, correspondences, review articles, in vitro studies, pediatric patient population, or animal studies, 37 articles remained for full-length review. Twenty-three of them were excluded from the full-length review as they did not report the outcome of interest, while nine articles were excluded because they were descriptive studies without comparative analysis.

Thus, the final analysis included five articles [19,20,21,22,23], including 1990 patients with left ventricular (LV) systolic dysfunction, defined as an LV ejection fraction (LVEF) of lower than 50%. Among the 5 RCTs which were enrolled for a qualitative review, 3 of them were further examined in meta-analysis for outcomes of interest, which included difference of time to sinus rhythm restoration and sinus rhythm restoration rate in AF patients with LV systolic dysfunction. The literature retrieval, review, and selection process are demonstrated in Figure 1. The characteristics and quality assessment of the included studies are presented in Table 1. An assessment of the risk of bias is shown in Figure 2.

### 3.1. Sinus Rhythm Restoration Rate in AF Patients with LV Systolic Dysfunction

There was a significant association between sinus rhythm restoration rate and ranolazine in AF patients with the pooled RR of 2.87 (95% CI, 2.48–3.32, Figure 3. There was a significant heterogeneity with I^2^ of 98%. The pooled estimated standard mean difference of time to sinus rhythm restoration of AF patients with LV systolic dysfunction was 2.46 h less in the ranolazine group compared to patients without ranolazine administration (95%CI: 2.27–2.64, I^2^ = 98%, Figure 4). There was a significant heterogeneity with I^2^ of 98%.

### 3.2. Incidence of AF in Various Clinical Settings

Two enrolled studies reported the effectiveness of ranolazine in the prevention of AF by reducing its episodes. Because of the small number of patients with adverse outcomes, pooled analysis was not calculated. Scirica et al. [21] reported a post hoc analysis of MERLIN-TIMI 36 trials, which showed the benefit of ranolazine for the prevention of AF in patients with non-ST segment elevation myocardial infarction (NSTEMI). Overall, patients in the ranolazine arm had a trend towards fewer episodes of AF [75 (2.4%) vs. 55 (1.7%) patients, *p*-value = 0.08] detected in continuous ECG monitoring during the first 7 days after randomization. Moreover, over a median 1-year follow-up, patients treated with ranolazine experienced fewer episodes of an event compared with the placebo group (2.9 vs. 4.1%, RR 0.71, *p*-value = 0.01). In contrast, Bekeith et al. [19] reported a trend of its benefit towards prevention of postoperative AF with a 38% reduction in the AF incidence. However, due to the small sample size this did not reach statistical significance (*p*-value = 0.530).

### 3.3. Safety Profile of Ranolazine

Four enrolled studies determined the safety profile of ranolazine [21]. The results concordantly supported that ranolazine, either being used alone or as an add-on therapy, was well-tolerated in the absence of documented serious side effects. Koskinas et al. [20] and Tsanaxidis et al. [23] reported that patients in the drug combination arm had more experience in mild adverse effects, including dizziness, nausea, and mild arterial hypotension which was gradually reversible without the need for treatment discontinuation. Koskinas et al. also reported a significant increase in QTc from baseline in the combination arm (24.4 ± 7.1 vs. 19.1 ± 7.3 ms in the combination arm and control arm, respectively). However, neither patient had excessively prolonged QTc, nor had documented torsades de pointes.

### 3.4. Evaluation for Publication Bias

Funnel plots (Appendix A) and Egger’s regression asymmetry tests were performed to evaluate for publication bias in analyses evaluating sinus rhythm restoration in AF patients with LV systolic dysfunction. There was no significant publication bias with a *p*-value = 0.70.

## 4. Discussion

To the best of our knowledge, this is the first comprehensive meta-analysis to evaluate the efficacy and safety of ranolazine in AF patients with LV systolic dysfunction. Previous RCTs and meta-analysis support the effectiveness of ranolazine in preventing new-onset AF and its potential for pharmacological cardioversion [12,14,24,25]. However, due to the limited data from available studies, the role of ranolazine in patients with LV systolic dysfunction was uncertain. For this reason, we conducted a comprehensive systematic review, which included RCTs that enrolled patients with LV systolic dysfunction, referring to those with LVEF of lower than 50% in the trials. Our study demonstrated that ranolazine as an add-on therapy to amiodarone potentiates and accelerates the conversion of AF to sinus rhythm in patients with LV systolic dysfunction. The time to sinus rhythm restoration was 2.46 h shorter in the ranolazine arm. Furthermore, ranolazine tends to have efficacy in the prevention of new-onset AF and tends to be safe in patients with LV systolic dysfunction.

Ranolazine is an anti-anginal drug which has additional anti-arrhythmic properties. The anti-arrhythmic effects are derived from its ability to inhibit late INa and rapid delayed rectifier potassium channel (IKr) in both atria and ventricles [11,26,27]. The inhibition of late INa reduces intracellular sodium concentration and the subsequent intracellular calcium overload. In contrast, the inhibition of IKr prolongs the refractory period. Moreover, at the atrial level, ranolazine also selectively inhibits peak Ina, which reduces atrial action potential upstroke, increases the diastolic threshold of excitation, and increases post-repolarization refractoriness. These electrophysiological actions show its role in AF suppression [11,28].

### 4.1. Ranolazine for Pharmacological Cardioversion of AF in LV Systolic Dysfunction

With respect to the pharmacological conversion of ranolazine, our meta-analysis included 3 RCTs. These trials enrolled patients with LV systolic dysfunction and showed that ranolazine was effective in pharmacological cardioversion of AF when being used as a combination therapy with intravenous amiodarone. The pooled results concordantly show that this combination significantly enhances the conversion rate of AF to sinus rhythm, and reduces AF conversion time when compared to amiodarone alone. Our findings are consistent with the data from a recent network meta-analysis performed by Tsiachris et al., which reported that amiodarone with an add-on of ranolazine was the most effective regimen for sinus rhythm restoration within 24 h [29]. Regarding the time to sinus rhythm restoration, this study found that flecainide and vernakalant arms had a faster effect. Nevertheless, flecainide and vernakalant had limitations in use among patients with structural heart disease as well as heart failure, the main clinical profile of our included study populations.

This synergistic effect of AF suppression results from an intrinsic ability of ranolazine to enhance amiodarone-mediated INa inhibition [30]. In short, ranolazine blocks INa in the activated state (use-dependent) [28,30], whereas amiodarone blocks it in the inactivated state. Moreover, both of them have the potentials to inhibit IKr. As a result, their combination potentially leads to a greater atrial-selective depression of the conduction velocity and a greater prolongation of the atrial refractory period. Because of these pharmacological properties, a higher AF suppression ability is achieved when both drugs are used together [30,31].

In addition to these electrophysiological effects, experimental findings of isolated atrial cells reported by Burashnikov et al. demonstrated that ranolazine provided a higher suppression of INa in the failing ones compared to those in normal atria [32]. This even better efficacy of ranolazine in failing atria is not unforeseen, because those failing or remodeled atria have an increased integral of late INa, which enhances ranolazine’s effects [33,34]. The remodeled atria in experimental studies may represent atria in patients with structural heart diseases, including those with LV systolic dysfunction. This hypothesis is supported by a randomized clinical trial conducted by Simopoulos et al., which reported that amiodarone alone or in combination with ranolazine converted post-operative atrial fibrillation faster in patients with reduced LVEF than in those with preserved LVEF [22].

### 4.2. Ranolazine for Prevention of AF in LV Systolic Dysfunction

According to the previous studies, the preventive role of ranolazine in new-onset AF is mostly derived from patients undergoing cardiac surgery [25,35]. One meta-analysis performed by Trivedi et al. also reported that ranolazine significantly lowered the incidence of post-operative AF (RR 0.44, 95% CI: 0.25–0.78, *p*-value = 0.005) [25]. However, the data on its potential to prevent the incidence of AF in settings outside post-operative circumstances are not well-established. Similarly, its use in patients with LV systolic dysfunction is not yet widely investigated.

In line with the conversion result, we identified 2 studies that revealed ranolazine’s ability in the prevention of AF in patients with LV systolic dysfunction. The potential of ranolazine to prevent AF in various clinical settings may be due to its ability to suppress trigger activity [11]. As mentioned above, failing hearts have conditions that favor the beneficial effects of ranolazine; this positive effect of ranolazine in LV systolic dysfunction patients is supported by growing evidence in various clinical settings, including those with post-operative AF patients [25]. Its use in this setting may therefore be considered, particularly in patients with LV systolic dysfunction who are not tolerant of or contraindicated to beta-blockers.

### 4.3. Ranolazine Safety in LV Systolic Dysfunction

For a ranolazine safety concern, a study showed no significant adverse events and proarrhythmia events between the ranolazine group and its control, even in patients with structural heart diseases [31]. This supports the safety profile of ranolazine in patients with LV systolic dysfunction. However, specific adverse events could not be analyzed in this study because of the limited data from the included randomized trials, which did not provide a separate report of each adverse event. 

Explanations for the drug’s safety may be due to the atrial-selective electrophysiological effects of ranolazine, as well as its additional antiarrhythmic actions on ventricular level. According to ranolazine’s ability to inhibit late INa in M-cells and Purkinje fibers, ranolazine protects failing hearts which have prolonged action potential duration from proarrhythmias by reducing transmural dispersion of repolarization and decreasing early afterdepolarization [11,26]. Experimental studies also demonstrate that ranolazine not only increases the ventricular fibrillation (VF) threshold in the normal heart but also during myocardial ischemia [27]. Moreover, as ranolazine is a use-dependent antiarrhythmic drug, it may inhibit peak INa more effectively during VF, a very rapid heart rate state [27]. Nevertheless, the antifibrillatory effect of ranolazine needs further support from clinical trials before moving closer to the point of routine clinical use.

### 4.4. Study Limitations

There are some limitations to our study. Firstly, there is heterogeneity among studies, which include patients with recent-onset AF, NSTEMI, and postoperative AF. However, in the opposite view, this may be useful for real-world data in employing ranolazine in various clinical settings. Secondly, for AF prevention outcomes, we could identify only 2 RCTs, which resulted in a relatively small number of patients with adverse effects, so pooled analysis could not be calculated. Moreover, these 2 trials included different study populations; hence, the interpretation of AF prevention outcomes should be made with caution. Thirdly, for ranolazine safety information, we could not analyze specific adverse events because of the limited data from the included randomized trials. Further RCTs are thus needed to confirm this result. Finally, safety data of long-term ranolazine therapy in patients with LV systolic dysfunction are limited. As a result, its use in long-term settings, such as aiming to maintain sinus rhythm after cardioversion or catheter ablation, also needs future randomized clinical trials.

## 5. Conclusions

In AF patients with LV systolic dysfunction, our meta-analysis suggests that ranolazine as an adjunctive therapy to amiodarone potentiates and accelerates the conversion of AF to sinus rhythm. Moreover, ranolazine demonstrates good safety profiles. Further studies of ranolazine in AF management, particularly its use for the prevention of new-onset AF in patients with LV systolic dysfunction, are warranted.

## Figures and Tables

**Figure 1 diseases-09-00031-f001:**
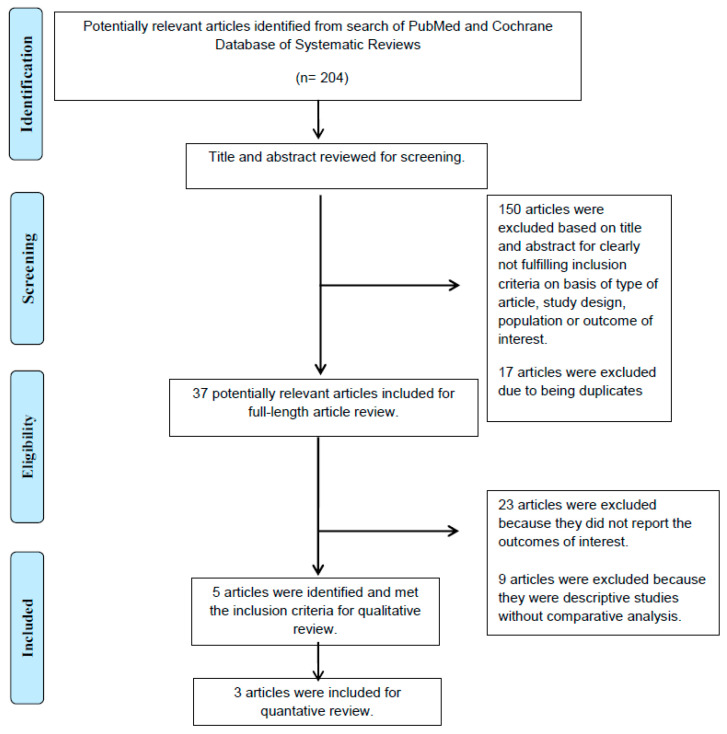
The literature retrieval, review, and selection process.

**Figure 2 diseases-09-00031-f002:**
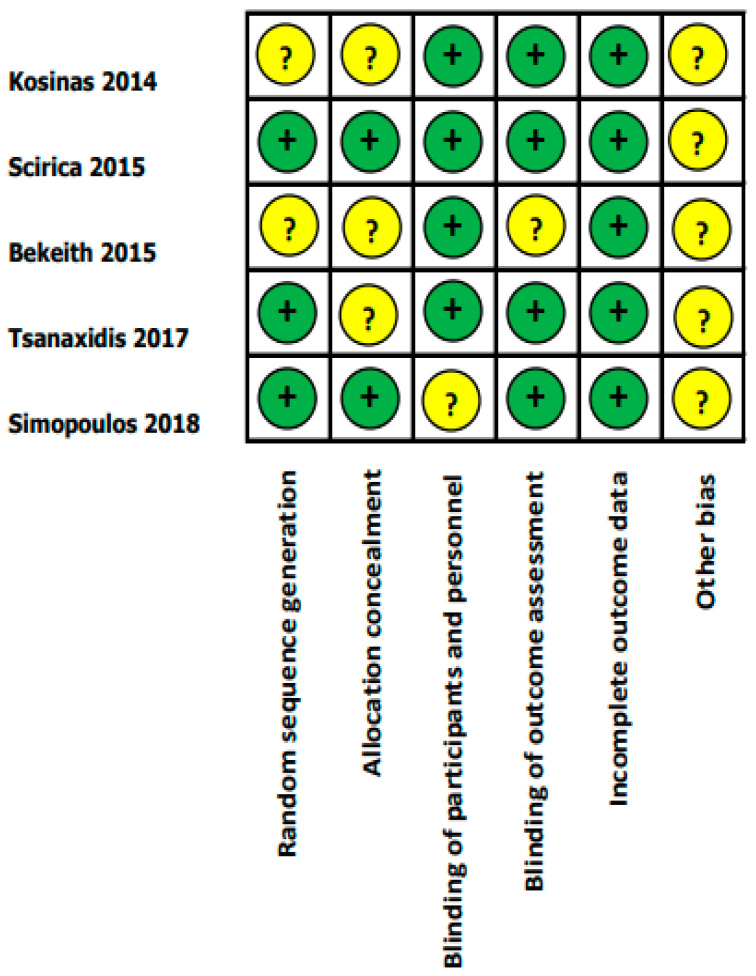
Cochrane risk of bias assessment of each included RCT. + indicates low risk of bias; ? indicates unclear risk of bias.

**Figure 3 diseases-09-00031-f003:**
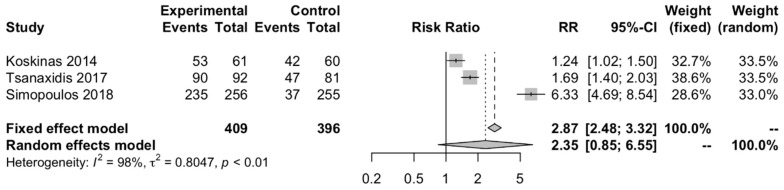
The forest plot meta-analysis showing the sinus rhythm restoration rate of ranolazine plus amiodarone group and amiodarone alone group.

**Figure 4 diseases-09-00031-f004:**
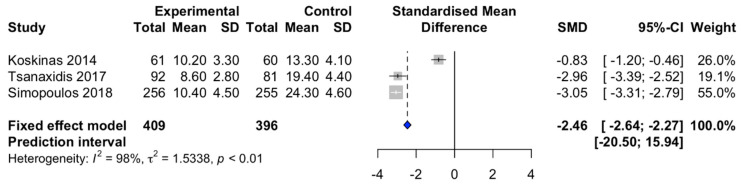
The forest plot meta-analysis showing sinus rhythm restoration time of the ranolazine plus amiodarone group and the amiodarone alone group.

**Table 1 diseases-09-00031-t001:** The characteristics and quality assessment of the included studies.

	Koskinas 2014 [20]	Scirica 2015 [21]	Bekeith 2015 [19]	Tsanaxidis 2017 [23]	Simopoulos 2018 [22]
Characteristics of RCT					
Design	Single center, RCT, SB	RCT, DB	Single center, RCT, DB	Single center, RCT, PROBE	Single center, RCT, SB
Intervention group	Ranolazine 1500 mg once plus IV amiodarone	IV ranolazine with a 200 mg bolus, then 80 mg/h for 12–96 h, then oral ranolazine 1000 mg bid	Oral ranolazine 1000 mg bid 48 h prior to surgery to 14th postoperative day	Oral ranolazine 1000 mg once plus IV amiodarone	Ranolazine 55 mg once, then 375 mg 6 h later, then 375 mg bid plus IV amiodarone
Control group	IV amiodarone loading dose 5 mg/kg then 50 mg/h for 24 h	Placebo	Placebo	IV amiodarone loading dose 5 mg/kg in 1 h then 50 mg/h	IV amiodarone 300 mg then 1125 mg/36 h
Primary endpoint, Number of events	Conversion of AF to SR within 24 h	Cardiovascular death, MI, recurrent ischemia	POAF	Time to conversion of AF	Time to conversion of POAF
Method of AF detection	Continuous ECG monitoring for 24 h	Continuous ECG monitoring for the first 7 days	Holter monitoring for 2 weeks	Continuous ECG monitoring	12-lead ECG every 4 h, if not, convert within 12 h, then Holter monitoring for 24 h
Follow-up period	24 h	12 months	2 weeks	24 h	36 h
Characteristics of patients in RCT					
Country	Greece	US	US	Greece	Greece
Study population	Recent-onset AF (<48 h)	NSTEMI, SR	Postoperative cardiac surgery, SR	Recent-onset AF (<48 h)	Postoperative CABG, POAF
No. of patients	61/60	3162/3189	27/27	92/81	256/255
Mean age (years)	66 ± 11/64 ± 9	63 ± 11/63 ± 11	64.3 ± 11.4	70 ± 10/67 ± 11	65.3 ± 9.5/65.5 ± 9.6
Male (%)	41/48	66.2/63.7	81	38/41	86.3/87.8
LA diameter (mm)	49 ± 8/46 ± 6	NA	NA	4.1 ± 0.4/4.2 ± 0.5	48.1 ± 2.7/48.3 ± 2.6
LVEF (%)	58 ± 7/54 ± 10 (LVEF < 50%, 25%/20%)	LVEF < 40%, 13.9%/13.4%)	45.4 ± 14.6	52 ± 10/53 ± 8 (LVEF < 50%, 14%/8%)	36.6 ± 4.8/36.5 ± 4.7
Study results					
Incidence of AF	NA	55/75	5/8	NA	NA
Conversion of AF to SR within 24 h	53/42	NA	NA	90/47	235/37
Time to AF conversion	10.2 + 3.3/13.3 + 4.1	NA	NA	8.6 ± 2.8/19.4 ± 4.4	10.4 ± 4.5/24.3 ± 4.6

Case/Control; AF = atrial fibrillation; bid = bis in die; CABG = coronary artery bypass graft; DB = double-blind; ECG = electrocardiography; h = hour; IV = intravenous; LA = left atrium; LVEF = left ventricular ejection fraction; mg = milligram; mm = millimeter; NA = not available; No. = number; NSTEMI = non-ST segment elevation myocardial infarction; POAF = post-operative atrial fibrillation; PROBE = prospective randomized open-label blinded end-point; RCT = randomized controlled trial; SB = single-blind; SR = sinus rhythm; US = United States.

## Data Availability

The data presented in this study are available in this article.

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
