# Peer review of "Role of Ranolazine in the Prevention and Treatment of Atrial Fibrillation in Patients with Left Ventricular Systolic Dysfunction: A Meta-Analysis of Randomized Clinical Trials"

_diseases, 2021, doi:10.3390/diseases9020031_

Round 1

Reviewer 1 Report

In this study Leelapatana P. et al. conducted a meta-analysis of published randomized controlled trials in order to investigate the efficacy and safety of Ranolazine in patients with atrial fibrillation (AF) and left ventricular systolic dysfunction.

Below find my comments:

  • In Figure 1, in the panel where literature search is described, it is specified that potentially relevant articles are identified from search of MEDLINE, EMBASE and Cochrane Database of Systematic Reviews, while in paragraph “Literature review and Search Strategy” in Material and Methods section as well as in Supplementary Data are cited as search database PubMed and Cochrane Database of Systematic Reviews. Please make the Figure consistent with the text.
  • In Table 1, explanation of the abbreviation LA used in “LA diameter” is missing.
  • In Results section, I suggest to dedicate a paragraph also to results related to sinus rhythm restoration time of AF patients as done for other results or, alternatively, to insert them in paragraph 3.1 as they are related to sinus rhythm restoration rate and data come from the same three article.
  • I can’t find Figure S3, referred to in paragraph 3.3, in Supplementary Data.

Author Response

Response to Reviewer#1

Comment #1

In Figure 1, in the panel where literature search is described, it is specified that potentially relevant articles are identified from search of MEDLINE, EMBASE and Cochrane Database of Systematic Reviews, while in paragraph “Literature review and Search Strategy” in Material and Methods section as well as in Supplementary Data are cited as search database PubMed and Cochrane Database of Systematic Reviews. Please make the Figure consistent with the text.

Response: We thank you for reviewing our manuscript. We had corrected the detail of literature search in each relevant panel, including Material and Methods section, Figure 1, as well as Supplementary Data.

Comment #2

In Table 1, explanation of the abbreviation LA used in “LA diameter” is missing.

Response: The abbreviation LA used in “LA diameter” was added in Table 1.

Comment #3

In Results section, I suggest to dedicate a paragraph also to results related to sinus rhythm restoration time of AF patients as done for other results or, alternatively, to insert them in paragraph 3.1 as they are related to sinus rhythm restoration rate and data come from the same three article.

Response: We agree with this comment. Theoretically, the longer time in AF, the higher risk of stroke occurrence. Therefore, this information should be emphasized in the distinct paragraph. We inserted results related to sinus rhythm restoration time of AF patients in paragraph 3.1 with the following text:

‘The pooled estimated standard mean difference of time to sinus rhythm restoration of AF patients with LV systolic dysfunction was 2.46 hours less in ranolazine group compared to patient without ranolazine (95%CI: 2.27-2.64, I2=98%, Figure 3). There was a significant heterogeneity with I2 of 98%.’

Comment #4

I can’t find Figure S3, referred to in paragraph 3.3, in Supplementary Data.

Response: We went over all corresponding figures in the text. The word “Figure S3” was changed to “Figure S1”. Figure S1 was also added to Supplementary Data.

We greatly appreciated the reviewer’s and editor’s time and comments to improve our manuscript. The manuscript has been improved considerably by the suggested revisions.

Reviewer 2 Report

I have read with great interest this meta-analysis of randomized controlled trials evaluating the effectiveness of ranolazine as add-on therapy to amiodarone in sinus rhythm restoration in atrial fibrillation patients with left ventricular systolic dysfunction. This is a generally well-written manuscript which adds valuable information about the role of ranolazine on AF cardioversion in patients with reduced ejection fraction. A nice interpretation of the results along with an insight into the mechanisms of action of ranolazine is also provided in the discussion. Nevertheless, there are some issues that need to be revised.

  1. Trials included in the analysis: As a matter of fact, only three out of the five presented RCTs evaluated the outcome of interest (mean difference of time to sinus rhythm restoration and sinus rhythm restoration rate in AF patients with systolic dysfunction) and thus included in the two meta-analyses performed. There was no pooled analysis of the outcomes of the other two trials (by Scirica et al. and Bekeith et al.). Although the authors recognize this limitation of their study, it should be clearly stated in the abstract and in the main text that this is a meta-analysis of 3 and not of 5 studies.
  2. Evaluation of safety profile was one the aims of this meta-analysis. In the abstract, it is reported that no significant adverse events or proarrhythmias in ranolazine group were identified. However, there are no safety outcomes provided in the results section of the main text.
  3. A conclusion regarding the effectiveness of ranolazine in the prevention of AF in patients with LV systolic dysfunction cannot be derived from the analysis performed in this study.
  4. In the abstract, the authors report in the conclusions that ranolazine seems to be effective in the prevention of new-onset AF, but this cannot be derived from the previously reported results.
  5. Protocol registration: The study does not have a protocol registration, even if it is not mandatory, its publication guarantees greater reliability in the research.
  6. Heterogeneity: In the two meta – analysis forest plots presented by the authors, I2 values of 98% are indicative of large amount of heterogeneity. Please correct the comment about the heterogeneity of the first meta-analysis (line 126) accordingly. A short presentation
  7. Regarding the main text structure, the paragraph referring to the results of the first meta-analysis should be titled, similarly to that referring to the second meta-analysis.
  8. In order to provide more support to the preventive role of ranolazine in new-onset AF, a reference to the results of the RCT performed by Tagarakis et al (Effect of ranolazine in preventing postoperative atrial fibrillation in patients undergoing coronary revascularization surgery. Curr Vasc Pharmacol. 2013) is recommended.
  9. Is there any evidence supporting a potential role for ranolazine in life-threatening arrhythmias i.e. electrical storm? Please make a short reference.
  10. With regards to pharmacological cardioversion of paroxysmal AF, the authors may wish to mention a recent meta-analysis by Tsiachris et al. (Pharmacologic Cardioversion in Patients with Paroxysmal Atrial Fibrillation: A Network Meta-Analysis).

Author Response

Response to Reviewer#2

Comment #1

Trials included in the analysis: As a matter of fact, only three out of the five presented RCTs evaluated the outcome of interest (mean difference of time to sinus rhythm restoration and sinus rhythm restoration rate in AF patients with systolic dysfunction) and thus included in the two meta-analyses performed. There was no pooled analysis of the outcomes of the other two trials (by Scirica et al. and Bekeith et al.). Although the authors recognize this limitation of their study, it should be clearly stated in the abstract and in the main text that this is a meta-analysis of 3 and not of 5 studies.

Response: We thank you for reviewing our manuscript. We really appreciated your input and found your suggestions very helpful. We agree with the reviewer regarding the discordance between the number RCT studies included in qualitative and quantitative analysis. We had corrected this issue and clearly stated that this is a meta-analysis of 3 RCT studies. The revision was performed in the abstract, the main text, as well as Figure 1. The following text has been revised in the Results section:

Abstract: ‘We initially identified 204 studies and finally retrieved 5 RCTs. Three studies were analyzed in meta-analysis.’

Main text: ‘Among the 5 RCTs which were enrolled for a qualitative review, 3 of them were further examined in meta-analysis for outcomes of interest, which included difference oftime to sinus rhythm restoration and sinus rhythm restoration rate in AF patients with LV systolic dysfunction .’

Comment #2

Evaluation of safety profile was one the aims of this meta-analysis. In the abstract, it is reported that no significant adverse events or proarrhythmias in ranolazine group were identified. However, there are no safety outcomes provided in the results section of the main text.

Response: We appreciated the reviewer input and found the comment very important. The safety outcomes were added in the results section of the main text as the reviewer had suggested. The following text has been added in the Results section in the paragraph 3.3:

‘3.3. Safety profile of ranolazine

Four enrolled studies determined the safety profile of ranolazine.(21) The results concordantly supported that ranolazine, either being used alone or as an add-on therapy, was well-tolerated in the absence of documented serious side effects. Koskinas et al. and Tsanaxidis et al. reported that patients in the drug combination arm had more experience in mild adverse effects, including dizziness, nausea, and mild arterial hypotension which was gradually reversible without the need for treatment discontinuation. Koskinas et al. also reported a significant increase in QTc from baseline in the combination arm (24.4±7.1 vs. 19.1±7.3 ms in the combination arm and control arm, respectively). However, neither patient had excessively prolonged QTc, nor had documented torsade de pointes.’

Comment #3

A conclusion regarding the effectiveness of ranolazine in the prevention of AF in patients with LV systolic dysfunction cannot be derived from the analysis performed in this study.

Response: We appreciated the reviewer input and found the comment very important. As meta-analysis could not be performed in the outcome regarding effectiveness of ranolazine for AF prevention in patients with LV systolic dysfunction, we agree that a conclusion in this outcome cannot be made. For the clarification, we had revised this in Conclusion section of both abstract and main text. The following text has been added in the Conclusion section:

Abstract: ‘In AF patients with LV systolic dysfunction, ranolazine as an add-on therapy to amiodarone potentiates and accelerates the conversion of AF to sinus rhythm. Moreover, ranolazine shows good safety profiles. Further studies to investigate the effectiveness of ranolazine in prevention of new-onset AF are needed.’

Main text: ‘In AF patients with LV systolic dysfunction, our meta-analysis suggests that ranolazine as an adjunctive therapy to amiodarone potentiates and accelerates the conversion of AF to sinus rhythm. Moreover, ranolazine demonstrates good safety profiles. Further studies of ranolazine in AF management, particularly its role for prevention of new-onset AF in patients with LV systolic dysfunction, are warranted.’  

Comment #4

In the abstract, the authors report in the conclusions that ranolazine seems to be effective in the prevention of new-onset AF, but this cannot be derived from the previously reported results.

Response: We appreciated the reviewer input and found the comment very important. The following text has been revised in the abstract:

‘Further studies of ranolazine in AF management, particularly its role for prevention of new-onset AF in patients with LV systolic dysfunction, are warranted.’

Comment #5

Protocol registration: The study does not have a protocol registration, even if it is not mandatory, its publication guarantees greater reliability in the research.

Response: Thank you very much for the important suggestion. We have added this point in the revised manuscript. The following text has been added in the Materials and Methods; literature review and Search Strategy part:

‘The protocol for this systematic review and meta-analysis was registered with PROSPERO (International Prospective Register of Systematic Reviews): CRD42020204769.’

Comment #6

Heterogeneity: In the two meta – analysis forest plots presented by the authors, I2 values of 98% are indicative of large amount of heterogeneity. Please correct the comment about the heterogeneity of the first meta-analysis (line 126) accordingly.

Response: We appreciated the reviewer input and sorry for the mistake. We have corrected this point in the revised manuscript. The following text has been changed in the result; part 3.1 sinus rhythm restoration rate in AF patients with LV systolic dysfunction:

‘The pooled estimated standard mean difference of time to sinus rhythm restoration of AF patients with LV systolic dysfunction was 2.46 hours less in ranolazine group compared to patient without ranolazine administration (95%CI: 2.27-2.64, I2=98%, Figure 3). There was a significant heterogeneity with I2 of 98%.’

Comment #7

Regarding the main text structure, the paragraph referring to the results of the first meta-analysis should be titled, similarly to that referring to the second meta-analysis.

Response: Thank you very much for your important suggestions. The paragraph was titled under part 3.1 “sinus rhythm restoration rate in AF patients with LV systolic dysfunction”.

Comment #8

In order to provide more support to the preventive role of ranolazine in new-onset AF, a reference to the results of the RCT performed by Tagarakis et al (Effect of ranolazine in preventing postoperative atrial fibrillation in patients undergoing coronary revascularization surgery. Curr Vasc Pharmacol. 2013) is recommended.          

Response: Thank you very much for the recommendation. Thus, we added the reference to the study performed by Tagarakis et al. as the reviewer’s suggestion. Moreover, we also additionally provided reference to the results of the meta-analysis performed by Trivedi et al. (Efficacy of ranolazine in preventing atrial fibrillation following cardiac surgery: Results from a meta-analysis. J Arrhythm. 2017). The following text has been provided in the discussion; part 4.2 Ranolazine for prevention of AF in LV systolic dysfunction:

‘According to the previous studies, the preventive role of ranolazine in new-onset AF is mostly derived from patients undergoing cardiac surgery (25,35). One meta-analysis performed by Trivedi et al. also reported that ranolazine significantly lowered incidence of post-operative AF (RR 0.44, 95% CI: 0.25-0.78, p-value = 0.005).(25) However, the data on its potential to prevent incident AF in settings outside post-operative circumstances are not well-established. Similarly, its use in patients with LV systolic dysfunction is not yet widely investigated.’

Comment #9

Is there any evidence supporting a potential role for ranolazine in life-threatening arrhythmias i.e. electrical storm? Please make a short reference.

Response: Thank you very much for the interesting point and comments. We added more discussion about the interesting evidence of the role of ranolazine in life-threatening arrhythmias, including ventricular fibrillation. The following text has been added in the discussion part:

‘Experimental studies also demonstrate that ranolazine not only increases ventricular fibrillation (VF) threshold in the normal heart, but also during myocardial ischemia. (36) Moreover, as ranolazine is a use-dependent antiarrhythmic drug, it may inhibit peak INa more effectively during VF, a very rapid heart rate state. (36)  Nevertheless, the antifibrillatory effect of ranolazine needs further support from clinical trials for moving closer to the point of routine clinical use.’

Comment #10

With regards to pharmacological cardioversion of paroxysmal AF, the authors may wish to mention a recent meta-analysis by Tsiachris et al. (Pharmacologic Cardioversion in Patients with Paroxysmal Atrial Fibrillation: A Network Meta-Analysis).    

Response: Thank you very much for the important suggestion. We had updated the potential relevant publications in the discussion part as the reviewer suggested. The following text has been added in the discussion part:

‘Our findings are consistent with the data from a recent network meta-analysis performed by Tsiachris et al. which reported that amiodarone with an add-on ranolazine was the most effective regimen for sinus rhythm restoration within 24 hours. (29) Regarding the time to sinus rhythm restoration, this study found that flecainide and vernakalant arms had a faster effect. Nevertheless, flecainide and vernakalant had limitations in using among patients with structural heart disease as well as heart failure, the main clinical profile of our included study populations.’

We greatly appreciated the reviewer’s and editor’s time and comments to improve our manuscript. The manuscript has been improved considerably by the suggested revisions.

Round 2

Reviewer 2 Report

The authors have sufficiently revised their paper, responding to the reviewers' comments.